# Hepatic-Modulatory Effects of Chicken Liver Hydrolysate-Based Supplement on Autophagy Regulation against Liver Fibrogenesis

**DOI:** 10.3390/antiox12020493

**Published:** 2023-02-15

**Authors:** Yi-Ling Lin, Chih-Ying Chen, Deng-Jye Yang, Yi-Hsieng Samuel Wu, Yue-Jia Lee, Yi-Chou Chen, Yi-Chen Chen

**Affiliations:** 1Department of Animal Science and Technology, National Taiwan University, Taipei City 10673, Taiwan; 2Institute of Food Safety and Health Risk Assessment, National Yang Ming Chiao Tung University, Taipei City 11221, Taiwan; 3Institute of Food Science and Technology, National Taiwan University, Taipei City 10617, Taiwan; 4Great Billion Biotech Co., Ltd., New Taipei City 23452, Taiwan; 5Master Program in Global Agriculture Technology and Genomic Science, International College, National Taiwan University, Taipei City 10617, Taiwan

**Keywords:** antioxidant/anti-inflammatory effects, autophagy, chicken liver hydrolysate-based supplement, liver fibrosis, thioacetamide

## Abstract

Chicken-liver hydrolysates (CLHs) have been characterized as performing several biofunctions by our team. This study aimed to investigate if a CLH-based supplement (GBHP01^TM^) can ameliorate liver fibrogenesis induced by thioacetamide (TAA) treatment. Our results showed that the TAA treatment caused lower body weight gains and enlarged livers, as well as higher serum ALT, AST, and ALP levels (*p* < 0.05). This liver inflammatory and fibrotic evidence was ameliorated (*p* < 0.05) by supplementing with GBHP01^TM^; this partially resulted from its antioxidant abilities, including decreased TBARS values but increased TEAC levels, reduced GSH contents and catalase/GPx activities in the livers of TAA-treated rats (*p* < 0.05). Additionally, fewer nodules were observed in the appearance of the livers of TAA-treated rats after supplementing with GBHP01^TM^. Similarly, supplementing GBHP01^TM^ decreased fibrotic scars and the fibrotic score in the livers of TAA-treated rats (*p* < 0.05). Moreover, the increased hepatic IL-6, IL-1β, and TNF-α levels after TAA treatment were also alleviated by supplementing with GBHP01^TM^ (*p* < 0.05). Meanwhile, GBHP01^TM^ could decrease the ratio of LC3B II/LC3B I, but upregulated P62 and Rab7 in the livers of TAA-treated rats (*p* < 0.05). Taking these results together, the CLH-based supplement (GBHP01^TM^) can be characterized as a natural agent against liver fibrogenesis.

## 1. Introduction

Liver cancer ranks third among cancer categories globally, and approximately 830,000 people die with liver cancer every year [1]. Similar results have also been observed in Taiwan, and liver-related diseases always exist among the leading causes of death [2]. The liver is a silent organ, and does not always causes alarm before the development of severe liver damage; hence, consciousness of hepatoprotection has recently been developed via lifestyle modifications and functional food supplementation. A nutrient or functional food ingredient that ameliorates liver fibrogenesis before the onset of irreversible liver cirrhosis could be a hot topic for food scientists.

Food-derived protein hydrolysates have been suggested to provide functional properties such as antioxidant [3], lipid-lowering [4] and antihypertensive [5] effects, etc. It has been reported that protein hydrolysates obtained from specific hydrolyzing conditions could provide biological functions which mainly result from their amino acid composition, and peptides with specific sequences and structures [6,7,8]. Protein hydrolyzation offers a feasible application of animal by-products which usually contain abundant proteins in their feathers, blood, viscera, and skin. Several previous studies have indicated that the hydrolysates of animal by-products provide antioxidant [3] and lipid-lowering effects [9].

Candidates for functional-food ingredients should be either effective or low in cost. Regarding the poultry industry in Taiwan, approximately 10,000 metric tons of chicken livers (2.5% based on broilers’ wt.) have been produced in recent years [10], and they are either used only as fertilizer or incinerated. Plenty of proteins in chicken livers could be hydrolyzed to produce functional hydrolysates which are characterized by their lipid-lowering [9] and antioxidant effects [3]. Our previous study indicated that functional chicken-liver hydrolysates (CLHs) produced under conditions of pepsin digestion for 2 h and at 37 °C reveal the best antioxidant ability [3]. In addition, CLHs did not only offer antioxidant ability in D-galactose-induced mice [3], but also provided lipid-lowering effects in high-fat diet-induced hamsters [9]. These benefits are attributed to the free amino-acid profile and bioactive dipeptides, i.e., carnosine and anserine, in CLHs. Carnosine (β-alanyl-L-histidine, CNS) and anserine (β-alanyl-N-histidine) are natural histidine-containing and meat-sourced dipeptides, especially broilers, which demonstrate several biological activities, such as antioxidative activities, pH buffering, complexing of dangerous carbonyl compounds, and anti-glycation activities on proteins, etc. [11]. In the view of the agro-cycle value and the functional properties of CLHs, this study aimed to evaluate whether a CLH-based daily supplement could alleviate liver fibrogenesis via a TAA-induced rat’s model.

## 2. Materials and Methods

### 2.1. Chicken-Liver Hydrolysate (CLH)-Based Supplement and Chemical

A CLH-based supplement capsule (GBHP01^TM^) containing 200 mg CLH powder and 20 mg hesperidin (HAYASHIBARA HESPERIDIN^TM^ S, Nagase Taiwan Co., Ltd., Taipei, Taiwan) per capsule was generously offered by Great Billion Biotech Co., Ltd., New Taipei City, Taiwan. The composition of free amino acids and imidazole-ring dipeptides in a GBHP01^TM^ capsule were sent for analysis using an Amino Acid Analyzer (Hitachi L8800 amino acid analyzer, Hitachi High-Technologies Co., Tokyo, Japan) in the Food Industry Research and Development Institute (Hsin-Chu City, Taiwan). The total contents of free essential and non-essential amino acids and imidazole-ring dipeptides are 13.10, 65.04, and 0.07 mg per capsule (average 650 mg), respectively, where the total free branched-chain amino acid (BCAA) and taurine content are 5.77 mg and 51.67 mg per capsule (Appendix A). Two capsules per day for a 60 kg adult is recommended as the daily intake by Great Billion Biotech. Co., Ltd. (New Taipei City, Taiwan). According to the dose translation between rats and humans [12], 133 and 266 mg GBHP01^TM^ content per kg BW were calculated as 1× and 2× doses, respectively, for rats in this study. All other chemicals used in this study are of analytical grade.

### 2.2. Animals and Treatments

Forty male Wistar rats at the age of 5 weeks were purchased from BioLASCO Taiwan Co., Ltd. (Taipei, Taiwan), and acclimatized for one week before commencement of the experiment. The environmental parameters of the animal room were 22 ± 2 °C and 12/12 h light/dark cycle; two rats with ear tags (No. 1 and 2) were housed per cage. All rats were randomly assigned to one of the following groups (n = 8 per group): (1) CON: saline (intraperitoneal injection, i.p.) + 1.5 mL pure H_2_O (oral gavage); (2) TAA: 100 mg TAA/kg BW (i.p.) + 1.5 mL pure H_2_O (oral gavage); (3) TAA + SIL: 100 mg TAA/kg BW (i.p.) + 150 mg silymarin/kg BW in 1.5 mL pure H_2_O (oral gavage); (4) TAA + GBHP01(1X): 100 mg TAA/kg BW (i.p.) + 133 mg GBHP01^TM^ content/kg BW in 1.5 mL pure H_2_O (oral gavage); (5) TAA + GBHP01(2X): 100 mg TAA/kg BW (i.p.) + 266 mg GBHP01^TM^ content/kg BW in 1.5 mL pure H_2_O (oral gavage). It was reported that silymarin could ameliorate livre fibrogenesis via decreasing endoplasmic reticulum stress-related gene expressions, inflammatory cytokines, collagen accumulation, and matrix metalloproteinase (MMP-2 and MMP-9) activities, as well as increasing tissue inhibitors of metalloproteinase gene expressions (TIMP-1 and TIMP-3) in livers [13]; hence, silymarin was applied as a positive control in this study. Saline and TAA were injected thrice per week (Monday, Wednesday, and Friday). Meanwhile, silymarin and GBHP01^TM^ were orally given daily. Water and diet (Laboratory Rodent Diet 5001, PMI Nutrition International/Purina Mills LLC, Richmond, IN, USA) were offered *ad libitum* throughout an 8-week experiment. The animal use and protocol were reviewed and approved by the National Taiwan University Care Committee (IACUC No.: NTU-110-EL-00140). The average daily feed and water intakes per cage were calculated as the volumes of diet and water consumption per rat on a daily basis, respectively. All rats were fasted overnight (8 h) before being euthanized by sodium pentobarbital injection in the final day of experiment. The heart, liver, and kidney from each rat were removed, weighed, and then stored at −80 °C for further analyses.

### 2.3. Determination of Serum Biochemical Values

Blood samples were collected via abdominal aorta and placed at room temperature for clotting, and then centrifuged at 3000× *g* and 4 °C for 10 min to obtain sera. Serum alanine aminotransferase (ALT), aspartate aminotransferase (AST), albumin (ALB), alkaline phosphatase (ALP), total triglyceride (TG), and total cholesterol (TC) were analyzed by using commercial enzymatic kits (Randox Laboratories Ltd., Crumlin, County Antrim and Moorgate, London, UK).

### 2.4. Determination of Antioxidant Capacities and Lipid Peroxidation, and Inflammatory Cytokines in Livers

First, the liver homogenate was prepared (0.5 g liver + 4.5 mL phosphate-buffered saline). After a centrifugation (12,000× *g*), supernatant was collected and stored at −80 °C for further analyses. Protein levels in supernatant were measured with a Bio-Rad protein assay kit (Cat. #: 500-0006, Bio-Rad Laboratories, Inc., Hercules, CA, USA). Thiobarbituric acid reactive substances (TBARS), trolox equivalent antioxidant capacity (TEAC), and reduced glutathione (GSH) levels as well as the activities of superoxide dismutase (SOD) and catalase (CAT) in livers were assayed according to the previous methods [14]. GPx activity was analyzed using a commercial kit (Ransel Glutathione Peroxidase, Randox Laboratories Ltd., Antrim, UK). The levels of inflammatory cytokines, i.e., interleukin-1β (IL-1β), IL-6, and tumor necrosis factor-alpha (TNF-α) were measured using an enzyme-linked immunosorbent assay (ELISA) and referenced according to the commercial manufacturer’s instructions (eBioscience Inc., San Diego, CA, USA).

### 2.5. Histopathological Examination

According to the methods of our previous report [15], histopathological examinations were performed by using hematoxylin and eosin (H&E) as well as Sirius red staining, respectively. The microscopic analyses were observed by a Zeiss Axioskop 340 microscope AxioCam ERc 5 s camera system with AxioVision Release 4.8.2 (06-2010) 341 software (Carl Zeiss Microscopy, LLC, Thornwood, NY, USA). Via a double-blind test, the histology activity index (HAI) scores [16] and Metavir scores [17] were applied to determine the levels of inflammatory and fibrotic status in the liver, respectively. Briefly, the HAI scores include portal inflammation (0, 1, 3, and 4 points), intralobular inflammation (0, 1, 3, and 4 points), and periportal necrosis (0, 1, 3, 4, 5, 6, and 10 points). The Metavir score evaluates the fibrotic severity (levels of collagen accumulation) of liver fibrosis, which is categorized into five levels dependent on the severity, from fibrosis to cirrhosis.

### 2.6. Western Blotting

The procedure of sample preparation, SDS-PAGE, transfer, and antibody hybridization were conducted according to a previous study [15]. Electrophoresis was performed by using Bio-Rad’s Mini-PROTEAN Tetra Cell System (BioRad Laboratories Inc., Hercules, CA, USA), and the proteins were transferred via the Criterion Wet Transfer Blotter. The antibodies used in this study were β-actin (1:5000 dilution; #3700, Cell Signaling Technology Inc., Danvers, NJ, USA), LC3B (1:1000 dilution; #2775, Cell Signaling Technology Inc.), P62 (1:5000 dilution; EPR4844, Abcam Co., San Francisco, CA, USA), and Rab7 (1:1000 dilution; #9367, Cell Signaling Technology Inc.). The secondary antibody was anti-mouse IgG-horseradish peroxidase (β-actin: 1:10,000 dilution; LC3B, P62, and Rab7: 1:3000 dilution; Thermo Fisher Scientific Inc., Waltham, MA, USA). The protein bands were detected with the enhanced chemiluminescence (ECL) kit (Immobilon^TM^ Western, Millipore Co., Billerica, MA, USA) under a MultiGel-21 (TOP BIO CO., New Taipei City, Taiwan), meanwhile the Image J (National Institutes of Health, Bethesda, MD, USA) was used to quantify the optical density of the protein bands, using the β-actin band as a reference. The folds of protein expressions of other groups were expressed relative to that of the CON group.

### 2.7. Statistical Analysis

The experiment was conducted using a completely randomized design (CRD). Data were analyzed by using SAS 9.4 software (SAS Institute Inc., Cary, NC, USA). When a significant difference (*p* < 0.05) among groups was detected by the one-way analysis of variance (ANOVA), differences among treatments were tested using the least significant difference (LSD) test. Liver HAI and Metavir scores (non-parametric data) between groups were assessed using Kruskal–Wallis tests and Dunn’s post hoc test.

## 3. Results

### 3.1. Effects of Chicken Liver Hydrolysate (CLH)-Based Supplement on Growth Performance, Liver Size, and Serum Biochemical Values of Rats

After 8 weeks of experiment, TAA treatment resulted in the lower body weight of rats (approximately a 12.19% reduction), while lower weight increases and smaller food intakes were observed in TAA-treated rats than in control rats (*p* < 0.05) (Table 1). No (*p* > 0.05) difference on water intakes was recorded among groups. With regard to organ sizes (Table 1), although the relative sizes of heart and kidney were not (*p* > 0.05) influenced by TAA treatment or silymarin/GBHP01^TM^ supplementation, TAA treatment enlarged (*p* < 0.05) the liver sizes of rats. This enlargement phenomenon was lessened (*p* < 0.05) by GBHP01^TM^ supplement. After sacrifice, the serum biochemical values were found as shown in Table 1. In comparison with the reference serum biochemical values, the levels of liver damage indices in sera, i.e., AST, ALT, and ALP, were significantly (*p* < 0.05) increased (*p* < 0.05) by TAA treatment, although serum ALT levels among groups were within the range of rats. However, silymarin or GBHP01^TM^ supplementation apparently reduced (*p* < 0.05) these values to the reference values, while some values were even similar to those of CON group [AST: TAA + GBHP01(2X) vs. CON, *p* > 0.05; ALT: TAA + SIL and TAA + GBHP01(2X) vs. CON, *p* > 0.05; ALP: TAA + GBHP01(1X) and TAA + GBHP01(2X) vs. CON, *p* > 0.05]. Additionally, serum ALB values were reduced (*p* < 0.05) by TAA treatment, except in the TAA + GBHP01(2X) group. Focusing on serum lipid levels, no (*p* > 0.05) differences on TC levels were observed among groups. The TAA group had the lowest (*p* < 0.05) serum TG levels among groups, but a reverse (*p* < 0.05) effect was observed in silymarin and GBHP01^TM^-supplemented groups, while TAA + SIL and TAA + GBHP01(2X) groups even showed similar (*p* > 0.05) serum TG levels to the CON group.

### 3.2. Effects of CLH-Based Supplement on Antioxidative Capacities and Inflammatory Cytokine Level in Livers of Rats

As the results in Table 2 show, the TAA group has significantly higher TBARS levels and lower TEAC levels in liver tissues than the CON group (TBARS: 44.44% increase; TEAC: 18.03% decrease) (*p* < 0.05), while the silymarin or GBHP01^TM^-supplemented group had similar (*p* > 0.05) TBARS to the CON group and no (*p* > 0.05) difference in TEAC levels among GBHP01^TM^-supplemented groups and the CON group. Besides, the decreased pattern of reduced GSH levels was observed in TAA-treated groups, but silymarin or GBHP01^TM^ supplementation allowed for maintenance of the reduced GSH level in TAA-treated rats, especially in the group receiving 2X doses of GBHP01^TM^ supplementation (*p* < 0.05). With regard to antioxidant enzymatic activities, SOD activities were not (*p* > 0.05) different among groups. The TAA group showed reduced (*p* < 0.05) CAT and GPx activities compared to CON group, while silymarin or GBHP01^TM^ supplementation avoided reducing CAT activities, showing a result similar (*p* > 0.05) even to that of the CON group; reversed (*p* < 0.05) effects on GPx activities were only observed in GBHP01^TM^ supplemented groups. In hepatic proinflammatory cytokines, the TAA group had higher (*p* < 0.05) hepatic IL-1β (31.26% increase, *p* < 0.05), IL-6 (105.82% increase, *p* < 0.05), and TNF-α (28.89% increase) than the CON group. GBHP01^TM^ supplementation reduced (*p* < 0.05) these increased hepatic inflammatory cytokines of TAA-treated rats to levels similar (*p* > 0.05) even to those of the CON group, but silymarin supplementation only reduced the hepatic IL-6 level.

### 3.3. Effects of CLH-Based Supplement on Gross and Histopathological Examination in Livers of Rats

According to the gross appearance of livers (Figure 1A), the liver appeared pale, nodular, or even hubnailed on the external surface in TAA group, while these phenomena were amended by silymarin or GBHP01^TM^ supplementation. Via H&E stained observation (Figure 1A), well-organized hepatocytes and intact cells were observed in the CON group. However, some necrotic cells (arrows) and a monocyte-aggregated region (arrowheads) around the central or portal veins (CV or PV) were observed in TAA-treated rats’ liver section, but silymarin or GBHP01^TM^ supplementation lessened these observations. Similarly, the scores of HAI (periportal necrosis, intralobular inflammation, and portal inflammation) according to the H&E stains were increased by TAA treatments, while significant reductions (*p* < 0.05) in all three HAI scores were observed in the GBHP01^TM^ supplementation group; however, silymarin only demonstrated a reduced pattern (Figure 1B). Meanwhile, only scores of portal inflammation in the TAA + GBHP01(1X) and TAA + GBHP01(2X) groups were similar to that of the CON group (Figure 1B). With regard to a specific stain on fibrotic tissues (Sirius red stain), apparent and thick fibrotic scars and collagen deposition around the CV and PV were observed in the TAA group, whereas silymarin or GBHP01^TM^ supplementation ameliorated those pathological changes (Figure 2A). The Metavir score echoed the observations from Sirius red stains wherein the TAA group had the highest (*p* < 0.05) score. Silymarin treatment did not (*p* > 0.05) attenuate the liver Metavir score in the TAA-treated rats. Treatment with 1X or 2X GBHP01 supplement decreased (*p* < 0.05) the scores in TAA-treated rats, although the scores were still higher (*p* < 0.5) than those of the CON group (Figure 2B).

### 3.4. Effects of CLH Based Supplement on Autophagy Modulation in Livers of Rats

With regard to autophagy modulation in livers of TAA-treated rats, the ratio of LC3B II/LC3B I of TAA group was almost 1.8 relative fold to that of CON group (*p* < 0.05), but GBHP01^TM^ supplement significantly downregulated (*p* < 0.05) this ratio, reaching levels even similar (*p* > 0.05) to those of CON (Figure 3A). Furthermore, the downstream factors P62 and Rab7 were downregulated (*p* < 0.05) by TAA treatment (Figure 3B,C). There was a tendency toward higher P62 and Rab7 protein expressions in the TAA + SIL group than in the TAA group. However, GBHP01^TM^ supplementation significantly upregulated (*p* < 0.05) these two proteins in livers of TAA-treated rats; meanwhile, the P62 and Rab7 protein expression in the livers of the TAA + GBHP01(1X) and TAA + GBHP01(2X) groups were reversed to similar levels to those of the CON group (*p* > 0.05).

## 4. Discussion

The liver is a silent organ and does not always cause alarm before the development of severe injury. It has been indicated that the metabolism and absorption of nutrients are affected in damaged livers, which could reduce food intake, thus lowering body weight [19]. Thioacetamide (TAA) often acts as hepatotoxic compound in rodent models [15,16,20,21,22], because the intermediates of TAA, i.e., thioacetamide-S-oxide (TASO) and thioacetamide-S or S-dioxide (TASO2), could cause centrilobular hepatic necrosis. It has also been reported that TAA treatment could result in higher serotonin (5-HT) in the brains of rats, thus probably leading to anorexia [23]. The lower food intake and body weight were also observed in TAA-treated rodents [15,24,25,26]. Hence, it was speculated that the lower body weight in TAA-treated rats could be the result of loss of appetite (and lower food intake). Moreover, previous reports mentioned that chronic liver damage could initiate alpha smooth muscle actin (αSMA) expression and cause collagen secretion (fibrotic scars); this, combined with the chronic inflammation, may thus lead to liver swelling and increased liver weight [16.20, 24]. It has also been reported that some natural agents such as silymarin [20], curcumin [21], and noni juice [13] could ameliorate liver fibrogenesis via enhancing antioxidants or downregulating inflammation and fibrosis-related proteins. The major ingredients in GBHP01^TM^ supplement are chicken liver hydrolysates (CLHs) and hesperidin, which are both characterized by antioxidant abilities [3,27]. Moreover, it was reported that CLHs show hepatoprotection, which is attributed to the free amino acid profile (BCAAs, taurine, aspartic acid, and glutamic acid) and imidazole-ring dipeptide (anserine) in CLHs [14,28], as well as the downregulation of hesperidin on nuclear factor-κB (NFκB), transforming growth factor-β (TGFβ) and connective tissue growth factor (CTGF) in CCl_4_-induced livers [27].

Both H&E and Sirius red stains are often applied to observe the level of liver fibrogenesis [14,15]. The HAI score was diagnosed based on morphological alterations of hepatocytes and inflammatory-cell aggregation [16], while the Metavir score [17] was assayed according to the levels and locations of fibrotic scars in the liver due to the staining of Type I and III collagen by Sirius red reagent [29]. The TAA treatment can significantly decrease the antioxidant ability and increase inflammatory/fibrogenic responses, thus worsening liver damage to fibrosis or even cirrhosis [13,15,16,20,21,24]. According to the HAI scores (Figure 1B), it was revealed that the TAA treatment destructs the membrane of hepatocytes, thus leading the AST, ALT, and ALP’s release into the extracellular liquid; meanwhile, the increased oxidative stress triggers inflammatory/fibrogenic responses, i.e., IL-1β, IL-6, and TNF-α (Table 2), as well as collagen accumulation (fibrotic scars) (Figure 2). Based on the above results, GBHP01^TM^ supplementation could ameliorate TAA intoxication, which decreases cell destruction and inflammatory/fibrogenic responses. Actually, it has been reported that CLHs could decrease the proinflammatory cytokine secretions, i.e., IL-1β, IL-6, and TNF-α in alcoholic damaged livers [28] or high-fat diet fed livers [14]. Additionally, the hepatoprotection of hesperidin against CCl_4_ is due to downregulating inflammatory/fibrogenic responses and stimulating anti-inflammatory cytokine secretion (IL10) [27]. According to the summary of the possible hepatoprotection of hesperidin [30], the reversed disturbance of antioxidant ability and anti-inflammatory/anti-apoptotic factor expressions are the possible mechanisms underlying the hepatoprotection of hesperidin. Hence, it is supposed that that ameliorative effects of GBHP01^TM^ supplement against liver fibrogenesis could result from increased antioxidant ability and decreased proinflammatory-cytokine production in TAA-treated livers. Recently, the role of autophagy in liver disease development is often discussed. The autophagy-lysosomal pathway is essential for cellular metabolic homeostasis through lysosome degradation of proteins, glycogen, lipids and organelles such as damaged mitochondria and excess endoplasmic reticulum (ER) to provide nutrients and biomolecular building blocks for cell survival [31]. Recently, delayed autophagosome clearance in Niemann–Pick type C disease [32] and increased formation of autophagosomes in viral hepatitis have been observed [31]. It was reported that an acute ethanol induction could inhibit the mammalian target of rapamycin (mTOR) complex I, activate unc-51-like autophagy, activating kinase I, and then upregulate Forkhead box O3, thus stimulating autophagy [33]. It was implied that proper autophagy capability in the liver may be crucial to reduce the detrimental effects of ethanol consumption. Moreover, Chao et al. (2018) proposed that an acceleration of autophagy via blocking mTOR activity or increasing levels of transcription factor EB (TFEB) can protect the liver from ethanol-induced damage [34]. Besides, Eid et al. (2013) observed that chronic ethanol consumption promotes the production of autophagy-related proteins, but autophagosomes are blockaded by impaired autolysosomes, thus prompting autophagy dysfunction [35]. This phenomenon might be due to decreased hepatic lysosome numbers and increased lysosomal pH, and impaired trafficking of lysosomal enzymes by chronic ethanol exposure which thus increases the number of autophagosomes [36,37]. LC3B and P62 expressions are the most often mentioned in order to judge autophagy initiation and autophagosome accumulation [38], while Rab7 plays an essential role in autolysosome production [39]. As the results in Figure 3 show, TAA treatment increased the ratio of LC3B II/LC3B I but decreased P62 protein, which causes autophagosome maturity; moreover, TAA treatment decreased Rab7 protein, which reduces autolysosome production. These phenomena indicate that the TAA treatment blocks the autophagy-lysosomal pathway in livers, thus worsening liver injury. Similarly, our previous study reported that a blockaded autophagy is observed in myocardia of long-term high-fat diet-fed mice (20 weeks), where there possibly exist an autophagosome accumulation and autolysosome immaturation [14]. Meanwhile, the increased inflammatory and fibrotic protein levels in the myocardia of long-term high-fat diet-fed mice were assayed. Moreover, BCAAs, especially L-leucine, modulated autophagy and the metabolic pathway via mTOR signal cascades [40]. Taurine also could upregulate P62 in the liver autophagy of offspring in gestational diabetic mellitus rats [41]. Similarly, it was observed that hesperidin also can decrease autophagosome accumulation in livers of chlorpyrifos-treated rats [42]. As explored in the literatures, GBHP01^TM^ supplementation could regulate the autophagy-lysosomal cascade in liver fibrogenesis of TAA-treated rats via decreasing autophagosome accumulation and promoting autolysosome maturation (Figure 3). Although CLH and hesperidin are the major components in GBHP01^TM^, the protective level of hesperidin against liver damage in rats (CCl_4_, thioacetamide, and acetaminophen) should be 200 mg hesperidin/kg BW [26,43,44]. Because of the doses of GBHP01^TM^ used in this study (133 and 266 mg/kg BW rat) and the hesperidin portion per GBHP01^TM^ capsule [20 mg in capsule (650 mg)], only 4.1 and 8.2 mg hesperidin/kg BW were given to rats. Hence, it was assumed that the amelioration of TAA-induced liver fibrosis by hesperidin can be considered negligible in this study. To summarize, this regulation probably mainly results from the bioactive ingredients in the CLHs of GBHP01^TM^ supplements (i.e., the free amino acid profile, Appendix A).

## 5. Conclusions

In this study, chicken liver hydrolysate (CLH)-based supplements (GBHP01^TM^) showed an ameliorating effect against liver fibrogenesis induced by TAA treatment (Figure 4). The hepatoprotective effects of the GBHP01^TM^ supplement were shown in the results of hepatic histological, proinflammatory cytokine and blood biochemical analyses. Increased oxidative stress in liver tissues induced by TAA treatment was also decreased by GBHP01^TM^ supplementation. With regard to autophagy modulation in livers, TAA treatment promoted early stage autophagy in the liver, but resulted in autophagosome accumulation due to immature autolysosome. However, GBHP01^TM^ supplementation could reverse the autophagy molecular mechanism which prevents autophagosome accumulation due to mature autolysosome formation. The evidence showed that the ameliorative outcomes of this CLH-based supplement (GBHP01^TM^) against liver fibrogenesis may be attributed to the bioactive ingredients in CLHs, including anserine, taurine, glutamic acid, aspartic acid, BCAAs, etc.

## Figures and Tables

**Figure 1 antioxidants-12-00493-f001:**
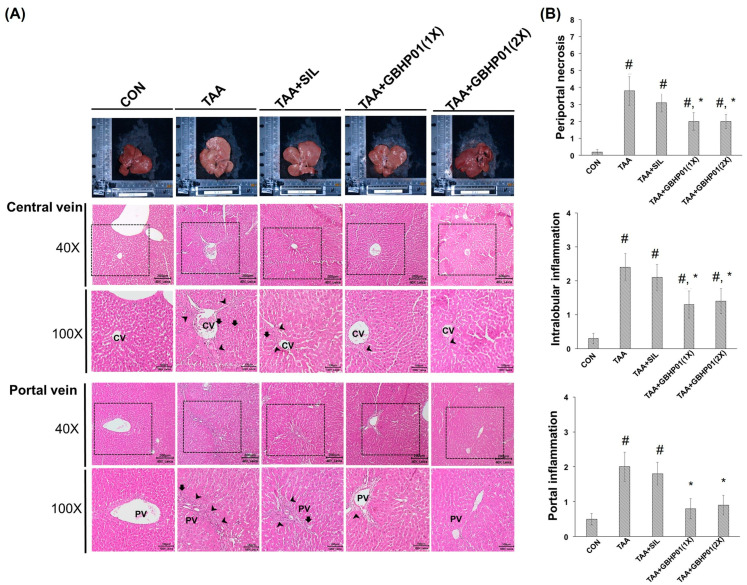
Effects of chicken liver hydrolysate-based supplement (GBHP01^TM^) on (**A**) the gross appearance and H&E stained illustrations of liver tissues of experimental rats (40× and 100×) and (**B**) the scores of HAI: periportal necrosis, intralobular inflammation, and portal inflammation. Data are given as mean ± SEM (n = 8). #: Signifcance against CON group (*p* < 0.05); *: Signifcance against TAA group (*p* < 0.05).

**Figure 2 antioxidants-12-00493-f002:**
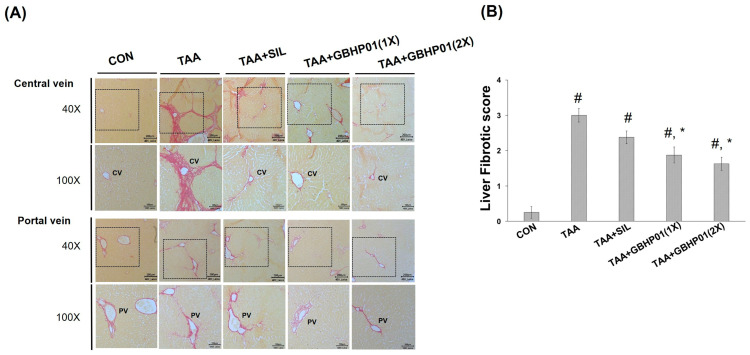
Effects of chicken liver hydrolysate-based supplement (GBHP01^TM^) on (**A**) Sirius red-stained illustrations of liver tissues of experimental rats (40× and 100×) and (**B**) the fibrotic scores according to Sirius red-stained observations of liver tissues. Data are given as mean ± SEM (n = 8). #: Signifcance against CON group (*p* < 0.05); *: Signifcance against TAA group (*p* < 0.05).

**Figure 3 antioxidants-12-00493-f003:**
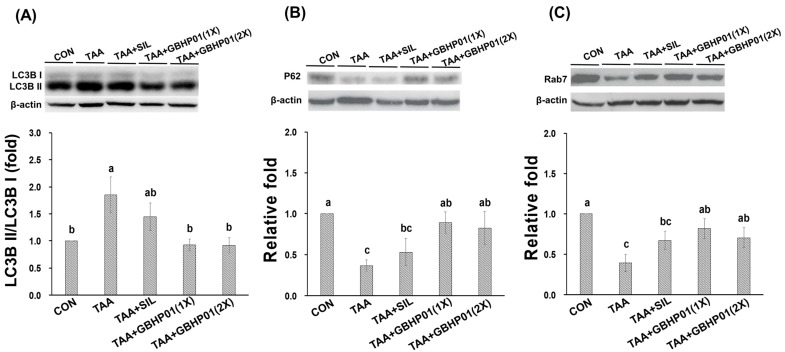
Effects of chicken liver hydrolysate-based supplement (GBHP01^TM^) on autophagy related protein expressions in livers of experimental rats: (**A**) LC3B II/LC3B I, (**B**) P62, and (**C**) Rab7. Data are given as mean ± SEM (n = 8). Data bars in each test parameter without a common letter represent a significant difference (*p* < 0.05).

**Figure 4 antioxidants-12-00493-f004:**
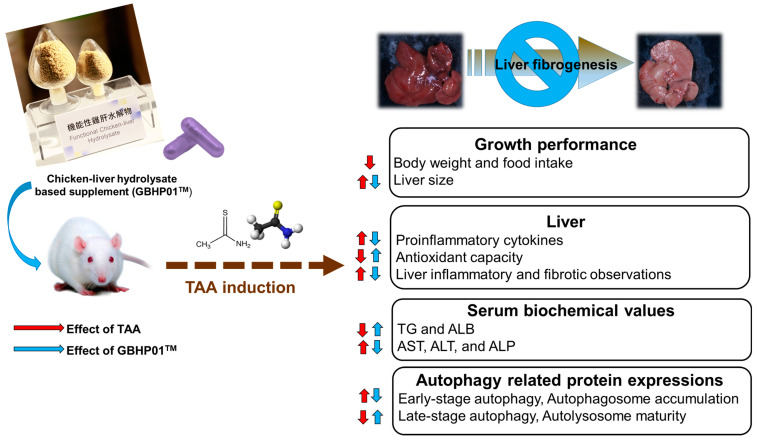
Schematic representation of liver fibrogenesis ameliorated by chicken liver hydrolysate-based supplement (GBHP01^TM^). The up arrow means an increased or upregulated effect; oppositely, the down arrow means a decreased or downregulated effect.

**Table 1 antioxidants-12-00493-t001:** Effects of chicken liver hydrolysate-based supplement (GBHP01^TM^) on growth performance, organ sizes, and serum biochemical values of experimental rats.

Group	CON	TAA	TAA + SIL	TAA + GBHP01(1X)	TAA + GBHP01(2X)
	*Growth performance*				
Initial body weight (g)	214.80 ± 2.01 a	212.6 ± 1.69 a	213.40 ± 2.17 a	215.35 ± 2.36 a	214.40 ± 2.07 a
Final body weight (g)	492.13 ± 8.68 a	425.25 ± 6.74 b	436.25 ± 6.51 b	438.13 ± 8.20 b	428.88 ± 9.41 b
Weight increase (g)	301.19 ± 7.56 a	211.56 ± 5.96 b	219.19 ± 5.48 b	230.38 ± 5.34 b	214.13 ± 5.34 b
Food intake (g/rat/day)	29.70 ± 0.84 a	25.51 ± 0.50 b	25.87 ± 0.68 b	25.69 ± 0.79 b	25.67 ± 0.81 b
Water intake (mL/rat/day)	58.02 ± 4.05 a	59.89 ± 2.80 a	58.08 ± 4.27 a	55.00 ± 2.95 a	57.82 ± 3.59 a
	*Size (g/100 g BW)*				
Heart	0.27 ± 0.01 a	0.27 ± 0.00 a	0.28 ± 0.01 a	0.28 ± 0.01 a	0.29 ± 0.01 a
Liver	3.06 ± 0.09 c	4.42 ± 0.14 a	4.20 ± 0.07 a	3.14 ± 0.05 bc	3.38 ± 0.12 b
Kidney	0.69 ± 0.02 a	0.72 ± 0.01 a	0.69 ± 0.01 a	0.72 ± 0.01 a	0.70 ± 0.01 a
	*Serum biochemical value*				
AST (U/L)	82.93 ± 4.62 c	135.02 ± 5.77 a	105.78 ± 4.51 b	99.09 ± 6.62 b	94.14 ± 3.79 bc
ALT (U/L)	23.57 ± 1.58 c	35.14 ± 1.26 a	28.01 ± 1.87 bc	28.37 ± 1.46 b	25.32 ± 1.66 bc
ALP (U/L)	90.05 ± 3.86 c	202.17 ± 8.45 a	159.16 ± 7.98 b	107.76 ± 5.59 c	100.17 ± 4.43 c
ALB (g/dL)	4.11 ± 0.18 a	3.53 ± 0.08 b	3.43 ± 0.22 b	3.63 ± 0.08 b	3.76 ± 0.17 ab
TC (mg/dL)	56.91 ± 2.18 a	61.35 ± 6.26 a	59.22 ± 2.45 a	57.62 ± 2.76 a	62.41 ± 3.52 a
TG (mg/dL)	66.10 ± 4.38 a	36.64 ± 2.53 c	57.88 ± 3.99 ab	50.11 ± 4.57 b	56.58 ± 2.48 ab

Data are given as mean ± SEM (n = 8, except food and water intakes, n = 4). Mean values in each test parameter without a common letter represent a significant difference (*p* < 0.05). Reference values of serum biochemical values of Wistar rats [18]: AST: 50~96 U/L; ALT: 24~49 U/L; ALP: 65~193 U/L; ALB: 3.4~4.8 g/dL; TC: 42.54~77.34 mg/dL; TG: 35.43~186.00 mg/dL.

**Table 2 antioxidants-12-00493-t002:** Effects of chicken liver hydrolysate-based supplement (GBHP01^TM^) on antioxidative status and proinflammatory cytokines in livers of experimental rats.

Group	CON	TAA	TAA + SIL	TAA + GBHP01(1X)	TAA + GBHP01(2X)
	*Antioxidant status*				
TBARS (nmole MDA eq./mg protein)	1.26 ± 0.09 b	1.82 ± 0.15 a	1.33 ± 0.13 b	1.37 ± 0.15 b	1.20 ± 0.09 b
Reduced GSH (nmole/mg protein)	60.54 ± 3.28 ab	50.03 ± 4.20 b	52.12 ± 1.65 b	57.71 ± 3.17 ab	64.57 ± 5.17 a
TEAC (μmole/mg protein)	1.22 ± 0.09 a	1.00 ± 0.07 c	1.02 ± 0.02 bc	1.17 ± 0.05 ab	1.13 ± 0.03 abc
SOD (unit/mg protein)	7.53 ± 0.67 a	6.39 ± 0.26 a	7.99 ± 0.38 a	8.06 ± 0.66 a	8.74 ± 0.79 a
CAT (unit/mg protein)	133.54 ± 8.01 a	80.60 ± 5.95 b	127.03 ± 7.08 a	132.48 ± 15.35 a	156.56 ± 13.38 a
GPx (unit/mg protein)	7.81 ± 0.22 ab	6.66 ± 0.42 c	7.01 ± 0.25 bc	8.76 ± 0.40 a	8.44 ± 0.45 a
	*Proinflammatory cytokine (pg/mg protein)*				
IL-1β	171.34 ± 13.36 bc	224.90 ± 5.91 a	211.86 ± 16.65 ab	145.29 ± 9.49 c	158.35 ± 12.04 c
IL-6	350.87 ± 22.08 c	722.15 ± 39.09 a	597.74 ± 41.45 b	424.46 ± 30.51 c	433.39 ± 30.61 c
TNF-α	80.06 ± 6.60 ab	103.19 ± 8.79 a	101.49 ± 4.98 a	67.92 ± 8.77 b	64.64 ± 7.25 b

Data are given as mean ± SEM (n = 8). Mean values in each test parameter without a common letter represent a significant difference (*p* < 0.05).

## Data Availability

Not applicable.

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
