# Peer review of "Hepatic-Modulatory Effects of Chicken Liver Hydrolysate-Based Supplement on Autophagy Regulation against Liver Fibrogenesis"

_antioxidants, 2023, doi:10.3390/antiox12020493_

Round 1

Reviewer 1 Report

The authors have tested the effect of administering a hydrolysate from chicken livers on TAA-induced fibrosis in rat livers. The effect of animal protein hydrolysate (it appears that chicken liver is used by convenience and availability) is of some interest, although the rather ill-defined composition makes it of limited broad interest. 

Specific comments

1. The study is funded by the company that produces the CLH and one author is an employee. They report no conflicts of interest, but it would be appropriate to address the exact relationship between the authors and the company, in particular any equity position.

2. The title and conclusions imply a mechanistic link between the effects of CLH and autophagy. While the results show that these area associated, they do not establish any mechanistic link.

3.  There are some issues with the statistics. In table 1 the liver weights are indicated to be different between control and TAA+GBHP01(1X). With the small effect size and listed error, I find this surprising. Overall, I would recommend rechecking statistics. In addition, in figures 1 and 2, injury or fibrosis scores are reported. Such scores are inherently nonparametric and should be analyzed via a nonparametric test.

4. The photos and histology in figure 1 are too poor a resolution to be evaluated.

5.  Correct the typo in line 91 (m should be mg).

6.  Land O'Lakes is the parent company of Purina which makes the rodent diet. Citing Purina would be more informative.

7. Line 327- proposing an hypothesis requires very little courage.

8.The up and down arrows in the graphical abstract are not informative.

9. Overall, the manuscript needs extensive editing for English usage and grammar.

Author Response

1. The study is funded by the company that produces the CLH and one author is an employee. They report no conflicts of interest, but it would be appropriate to address the exact relationship between the authors and the company, in particular any equity position.

Authors’ answer: Although one author in this manuscript (Mr. Yi-Chou Chen) is the director of Research and Development in Great Billion Biotech, Co., Ltd., he formulated GBHP01TM capsule (resource part), offered partial grant to this study (funding acquisition part), and reviewed the manuscript (writing—review and editing part).  Hence, we all believe that Mr. Yi-Chou Chen should deserve an authorship in this study.  With regard to the conflicts of interest, the roles of Dr. Yi-Chen Chen (corresponding author) and Mr. Yi-Chou Chen have been indicated in the revised part of Conflicts of Interest.

2. The title and conclusions imply a mechanistic link between the effects of CLH and autophagy. While the results show that the area associated, they do not establish any mechanistic link.

Authors’ answer: As we know, the autophagy-lysosomal pathway is essential for cellular metabolic homeostasis.  Participating proteins could be classified into three functional groups: (i) detectors of external factors and initiators of the autophagy process, e.g., ATG1 and mTOR; (ii) structural proteins for primary membrane and carrier proteins for degrading materials, e.g. P62, ATG8, and LC3B II; and (iii) markers for matured autolysosome, e.g., Rab7.  As the results in Figure 3, TAA treatment increased a ratio LC3B II/LC3B I but decreased P62 protein which cause autophagosome formation; moreover, decreased Rab7 protein which reduces autolysosome degradation.  Those phenomena indicate that the TAA treatment blocks the late-stage of autophagy-lysosomal pathway in livers, thus worsening the liver injury.  GBHP01TM supplement rescue the autophagy blockade in TAA-induced liver fibrogenesis via decreasing autophagosome accumulation and promoting autolysosome degradation.  Moreover, it was reported that leucine and taurine could accelerate the autophagy cascade (Zhang et al., 2007, Diabetes, 56:1647-1654; Luo et al., 2020, Life Science, 257: 117889).  Leucine (2.60 mg/capsule) and taurine (51.67 mg/capsule) are major free amino acids in GBHP01TM (Suppl. Table 1).  Hence, it is reasonably supposed that the amelioration of GBHP01TM against TAA-induced liver fibrogenesis is highly related to regulation of autophagy-lysosomal pathway.

3. There are some issues with the statistics. In table 1 the liver weights are indicated to be different between control and TAA+GBHP01(1X). With the small effect size and listed error, I find this surprising. Overall, I would recommend rechecking statistics. In addition, in Figures 1 and 2, injury or fibrosis scores are reported. Such scores are inherently nonparametric and should be analyzed via a nonparametric test.

Authors’ answer: Sorry for the typo in Table 1.  There is no difference on the liver weight between Control and TAA+GBHP01(1X), but a significant difference (p<0.05) on liver weights between Control and TAA+GBHP01(2X).  Hence, the statistical symbols on data of liver weights have been corrected in Table 1.  After we consult with a statistician, the liver injury or fibrotic scores have been statistically analyzed by using Kruskal-Wallis tests and Dunn's post hoc test.  Meanwhile, the Figure 1B and 2B, as well as the descriptions of liver HAI and Metavir scores in Section 3.3 have been modified accordingly.

4. The photos and histology in figure 1 are too poor a resolution to be evaluated.

Authors’ answer: The Figure 1 has been modified.  We hope the current resolution of Figure 1 is clear enough.

5. Correct the typo in line 91 (m should be mg).

Authors’ answer: Sorry for that typo.  It has been corrected in Ln 114.

6. Land O'Lakes is the parent company of Purina which makes the rodent diet. Citing Purina would be more informative.

Authors’ answer: Thanks for the reviewer’s friendly reminder.  The change has been made in Ln 126-127.

7. Line 327- proposing an hypothesis requires very little courage.

Authors’ answer: It has been observed the delayed autophagosome clearance and increased formation of autophagosomes in several diseases (Qian et al., 2021, Molecular Aspects of Medicine, 82:100973).  Although a chronic alcohol consumption promotes autophagy, chronic ethanol exposure decreased hepatic lysosome numbers and increased lysosomal pH and impaired trafficking of lysosomal enzymes, which increase number of autophagosomes (Qian et al., 2021, Molecular Aspects of Medicine, 82:100973).  The possible hypothesis or related references have been added in Ln 361-379.

8. The up and down arrows in the graphical abstract are not informative.

Authors’ answer: With regard to the reviewer’s concern, the information of up and down arrows has been offered in the legend of Figure 4.

9. Overall, the manuscript needs extensive editing for English usage and grammar.

Authors’ answer: Thanks for reviewer’s suggestion.  We have asked for an English editing service on our manuscript.  The English editing certificate is shown below. 

English Editing Cetificate

Reviewer 2 Report

After reading accuratedly the manuscript entitled "Hepatic-modulatory Effects of Chicken-liver Hydrolysate Based Supplement on Autophagy Regulation against Liver Fibrogenesis" I believe it reflexes a smart design of the study, releasing clear and sound results. However, before its full acceptance I suggest minor modification to its present form. Next, I will expose these suggested modifications: 

1.- Hesperidin should be displayed within the tittle, abstract, graphical abstract and introduction. These sentions do not make any reference to the herperidin despite it is one of the two GBHP01 components with biological activity.

2.- In line 78, the acronym BCAA is written for the first time without being defined. I must be defined.

3.- In line 91, "(2) TAA: 100 m TAA/kg BW (i.p.)" 100 m should be 100 mg. Right?

4.- In line 93, silymarin appears for the first time in the manuscript. It confused me because I did not know the properties. Maybe one or two lines, as lines 288-290 to introduce these compound to the reader could be helpful.

5.- In the Table 1, reference values for serum parametres (ALT, AST, ALP, TG, ALB, TC) should be included. Reading the range of normality near the parametre orientates the reader.

6.-In Table 1, I suppose that "Heat" is a mistyped word of "Heart". It should be corrected.

7.- Figure 1 and 2 quality should be strongly improved. 

8.- In line 323, mTOR shoud be defined inside the brackets after "mammalian target of rapamycin". Given that in line 328 it is written without a previous definition.

9.- In the graphical abstract, Figure 4, once GBHP01 has been defined its name should be used for the arrows legend. TAA should be defined in Figure 4 too.

Author Response

After reading accuratedly the manuscript entitled "Hepatic-modulatory Effects of Chicken-liver Hydrolysate Based Supplement on Autophagy Regulation against Liver Fibrogenesis" I believe it reflexes a smart design of the study, releasing clear and sound results. However, before its full acceptance I suggest minor modification to its present form. Next, I will expose these suggested modifications: 

1.- Hesperidin should be displayed within the tittle, abstract, graphical abstract and introduction. These sections do not make any reference to the hesperidin despite it is one of the two GBHP01 components with biological activity.

Authors’ answer: It was mentioned that the protective level of hesperidin against liver damage in rats (CCl4, thioacetamide, and acetaminophen) should be 200 mg hesperidin/kg BW rats (Perez-Vargas et al., 2014. Pharmacology 94: 80-89; Abo El-Magd, 2023. Life Sciences, 313: 121280; Ahmad et al., 2012. Toxicology Letters, 208:149-161).  Because of the doses of GBHP01TM used in this study (133 and 266 mg/kg BW rat) and the hesperidin portion per GBHP01TM capsule [20 mg in capsule (650 mg)], only 4.1 and 8.2 mg hesperidin/kg BW were supplemented to rats in this study.  Hence, it was assumed that the amelioration of hesperidin against TAA-induced liver fibrosis can be considered negligible in this study.  Due to this reason, we would like to keep the same title, abstract, graphic abstract, and introduction in our manuscript.

2.- In line 78, the acronym BCAA is written for the first time without being defined. It must be defined.

Authors’ answer: Thanks for the reviewer’s suggestion.  The full name of BCAA has been offered in Ln 100-101.

3.- In line 91, "(2) TAA: 100 m TAA/kg BW (i.p.)" 100 m should be 100 mg. Right?

Authors’ answer: Sorry for that typo.  It has been corrected in Ln 114.

4.- In line 93, silymarin appears for the first time in the manuscript. It confused me because I did not know the properties. Maybe one or two lines, as lines 288-290 to introduce these compounds to the reader could be helpful.

Authors’ answer: We concurred with the reviewer’s suggestion.  The purpose of silymarin application in this study has been offered in Ln 119-124.

5.- In the Table 1, reference values for serum parameters (ALT, AST, ALP, TG, ALB, TC) should be included. Reading the range of normality near the parameter orientates the reader.

Authors’ answer: The reference serum biochemical values of Wistart rats have been added in the footnote of Table 1.  With regard to liver damage indices, i.e. AST, ALT, and ALP, TAA treatment indeed increased these values of rats although serum ALT values among groups were within the range of reference values.  The description has been modified in Ln 208-215.

6.-In Table 1, I suppose that "Heat" is a mistyped word of "Heart". It should be corrected.

Authors’ answer: Sorry for that typo.  It has been corrected in Table 1.

7.- Figure 1 and 2 quality should be strongly improved. 

Authors’ answer: The Figure 1 and 2 have been modified.  We hope the current resolutions of Figure 1 and 2 are clear enough.

8.- In line 323, mTOR should be defined inside the brackets after "mammalian target of rapamycin". Given that in line 328 it is written without a previous definition.

Authors’ answer: Thanks for the reviewer’s suggestion.  The abbreviation of mammalian target of rapamycin has been added inside the brackets in Ln 367-368.

9.- In the graphical abstract, Figure 4, once GBHP01 has been defined its name should be used for the arrow legend. TAA should be defined in Figure 4 too.

Authors’ answer: Thanks for the reviewer’s suggestion. The Figure 4 and its legend has been modified.

Round 2

Reviewer 1 Report

The authors have made revisions in response to the previous review. There are only a few minor issues that have not been corrected. 

1.  Regarding conflict of interest, please state whether any authors have equity interest in Great Billion Biotech. It is ok if the answer is yes, it just needs to be disclosed. If there is no equity interest, please state that. Also please revise the position of Mr Chen to Director of Research and Development.

2. In line 372, please delete the word courageously.

Author Response

1. Regarding conflict of interest, please state whether any authors have equity interest in Great Billion Biotech. It is ok if the answer is yes, it just needs to be disclosed. If there is no equity interest, please state that. Also please revise the position of Mr Chen to Director of Research and Development.

Authors’ answer: Besides corresponding author and Mr. Yi-Chou Chen, other authors do not have any equity interest in Great Billion Biotech.  The statement has been added in the section of Conflicts of Interest in Ln 446-447.  We are sorry for the typo, the word of “direct” has been corrected to “director” in Ln 445.

2. In line 372, please delete the word courageously.

Authors’ answer: We concurred with the reviewer’s suggestion.  The word of “courageously” has been deleted in Ln 371.